# Adipose Tissue—Breast Cancer Crosstalk Leads to Increased Tumor Lipogenesis Associated with Enhanced Tumor Growth

**DOI:** 10.3390/ijms222111881

**Published:** 2021-11-02

**Authors:** Peter Micallef, Yanling Wu, Marco Bauzá-Thorbrügge, Belén Chanclón, Milica Vujičić, Eduard Peris, C. Joakim Ek, Ingrid Wernstedt Asterholm

**Affiliations:** Department of Physiology, Institute of Neuroscience and Physiology, The Sahlgrenska Academy at University of Gothenburg, Box 432, SE-405 30 Göteborg, Sweden; Peter.Micallef@gu.se (P.M.); Yanling.Wu@neuro.gu.se (Y.W.); marcos.bauza.thorbrugge@gu.se (M.B.-T.); belenchanclon@gmail.com (B.C.); milica.vujicic@gu.se (M.V.); Eduard.peris@wlab.gu.se (E.P.); joakim.ek@neuro.gu.se (C.J.E.)

**Keywords:** E0771, luminal B, breast cancer, adipose tissue, mouse, lipolysis, lipogenesis, adipocytes

## Abstract

We sought to identify therapeutic targets for breast cancer by investigating the metabolic symbiosis between breast cancer and adipose tissue. To this end, we compared orthotopic E0771 breast cancer tumors that were in direct contact with adipose tissue with ectopic E0771 tumors in mice. Orthotopic tumors grew faster and displayed increased de novo lipogenesis compared to ectopic tumors. Adipocytes release large amounts of lactate, and we found that both lactate pretreatment and adipose tissue co-culture augmented de novo lipogenesis in E0771 cells. Continuous treatment with the selective FASN inhibitor Fasnall dose-dependently decreased the E0771 viability in vitro. However, daily Fasnall injections were effective only in 50% of the tumors, while the other 50% displayed accelerated growth. These opposing effects of Fasnall in vivo was recapitulated in vitro; intermittent Fasnall treatment increased the E0771 viability at lower concentrations and suppressed the viability at higher concentrations. In conclusion, our data suggest that adipose tissue enhances tumor growth by stimulating lipogenesis. However, targeting lipogenesis alone can be deleterious. To circumvent the tumor’s ability to adapt to treatment, we therefore believe that it is necessary to apply an aggressive treatment, preferably targeting several metabolic pathways simultaneously, together with conventional therapy.

## 1. Introduction

Breast cancer is a common cause of death among women, but earlier detection and better treatments have greatly improved the chances of survival [1]. However, the prognosis is dependent both on the stage and on the molecular subtype. There are four main molecular subtypes of breast cancer: (1) luminal A (estrogen receptor-positive and/or progesterone receptor-positive (ER+ and/or PR+) and negative for human epidermal growth factor receptor 2 (HER2−)), (2) luminal B (ER+ and/or PR+/HER2+), (3) HER2-enriched (ER− and PR−/HER2+) and (4) triple-negative (ER−, PR− and HER2−). Luminal A breast cancers are the most common, and these cancers typically respond well to hormonal therapy and show the best prognosis [2]. In contrast, triple-negative cancers show the worst prognosis, but luminal B and HER2-enriched subtypes have also been associated with unfavorable prognosis [2,3,4]. Thus, there is still a need for new treatments to further improve the outcomes for women with breast cancer.

A growing body of research shows that adipose tissue enhances tumor progression in a paracrine and endocrine manner through many different plausible mechanisms [5,6,7,8,9]. For instance, the adipocyte-derived hormones leptin and adiponectin have been shown to affect tumor progression [10,11]. Moreover, intra-abdominal tumors often metastasize in an adipocyte-rich environment, at least partly because these cancer cells thrive on fatty acids released from adipocytes [12,13]. Breast cancers grow closely to mammary adipose tissue, implying that such lipid-related metabolic symbiosis may be of particular importance for these cancers. Indeed, a recent study shows that adipose tissue–breast cancer crosstalk leads to the transfer of lipids from adipocytes to the cancer cells, which in turn exhibit increased uncoupled fatty acid oxidation associated with increased tumor aggressiveness [14]. Besides this mechanism, fatty acids are also important for cell proliferation and tumor progression by serving as building blocks for membrane synthesis and acting as signaling molecules [15,16]. There are two ways in which a proliferating cell can meet its increase in fatty acid demand: either increase its fatty acid uptake or increase its de novo lipogenesis. In many cancers, de novo lipogenesis is markedly upregulated. In fact, enhanced de novo lipogenesis is recognized as one of the hallmarks of cancer metabolism (along with, e.g., increased glycolysis) and is associated with increased tumor progression, poor prognosis and resistance to chemotherapy [17,18,19,20,21]. Therefore, several de novo lipogenesis inhibitors have been developed as potential anti-cancer therapeutics. Many of these inhibitors appear to be rather promising and typically lead to reduced proliferation and increased sensitivity to chemotherapy in experimental cancer models [16]. As for breast cancer progression specifically, several studies show that fatty acid synthase (FASN) inhibition decreases proliferation and increases the chemosensitivity of many breast cancer cell lines as well as in experimental breast cancer models [22,23,24,25,26,27]. However, breast cancers are heterogenous with respect to fatty acid sensitivity [28] and metabolism, and, based on gene expression data, the less aggressive luminal A ER-positive breast cancer subtypes appear to rely on a balance between de novo lipogenesis and fatty acid oxidation as sources for both biomass and energy requirements, while basal-like, triple-negative subtypes may, to a larger extent, use exogenous fatty acids [29].

In this study, we exploited the luminal B breast cancer cell line E0771 [30,31] to further test the hypothesis that breast cancer tumors growing in close proximity to adipose tissue progress faster due to an altered fatty acid metabolism triggered by paracrine crosstalk and/or metabolic exchange between adipose tissue and cancer cells. Our findings confirm that adipose tissue accelerates tumor growth. Furthermore, we show that this adipose tissue-mediated increase in tumor growth is associated with increased adipocyte lipolysis and increased cancer cell lipogenesis in mice and in vitro. Monotherapy with lipolysis inhibitors had, however, no effect on tumor growth, while the lipogenesis inhibitor Fasnall showed more potential, although the dosage regimen appears critically important; our data suggest that intermittent (as opposed to continuous) treatment with a too low dose can enhance tumor growth.

## 2. Results

### 2.1. Adipose Tissue Accelerated E0771 Breast Cancer Growth In Vivo

E0771 breast cancer tumors growing in direct contact with adipose tissue (orthotopically) were 4.5-fold larger than ectopically growing tumors in chow-fed female mice, as judged by tumor weight captured at two weeks after E0771 cell injection (Figure 1A). In metabolically challenged high-fat-diet (HFD)-fed female mice that displayed increased body weight and glucose intolerance (Figure 1B,C), the orthotopic tumors were three-fold larger than ectopic tumors and orthotopic tumors grew two-fold larger than those of the healthy chow-fed mice. The ectopic tumors were three-fold larger in HFD-fed than in chow-fed mice, although this difference did not reach statistical significance (*p* = 0.06) (Figure 1A). Thus, mammary adipose tissue is a tumor-promoting microenvironment in both healthy lean and HFD-fed obese mice.

### 2.2. E0771 Breast Cancer Induced Increased Lipolysis, but Lipolysis Inhibitor Treatment Had No Effect on E0771 Tumor Growth In Vivo

The breast cancer cell lines MCF-1 and MDA MB 231 stimulated lipolysis in 3T3-L1 adipocytes, and the resultant elevation of fatty acids led to increased breast cancer fatty acid metabolism associated with increased proliferation and migration in vitro [32]. We thus hypothesized that the enhanced growth of orthotopically injected E0771 cells, at least in part, relies on increased lipolysis leading to increased fatty acids levels—especially locally, in the extracellular compartment within adipose tissue. To test this hypothesis, we first analyzed the ability of E0771 cancer cells to induce lipolysis in vitro. To this aim, 3T3-L1 adipocytes were treated for 24 h with conditioned media from E0771, ER+/PR+ luminal A (MCF7 and T47D), HER2+ (SKBR3) and triple-negative cell lines (MDA MB 231 and CAL-120). The triple-negative cell lines and E0771 were able to induce lipolysis, although the effect was rather small, while the other cancer cell lines were unable to induce lipolysis under these experimental conditions (Figure 2A). In vivo, E0771-bearing mice (orthotopically injected) did not become cachectic, as judged by their unchanged body and mammary adipose tissue weight (Figure 2B). Nevertheless, their fatty acid levels were increased, indicating increased lipolysis, and this increase could be prevented by treatment with the lipolysis inhibitor Acipimox (Figure 2C). Contrary to our hypothesis, Acipimox had no effect on tumor growth either in lean or in HFD-induced obese mice (Figure 2D). Similarly, Atglistatin, another lipolysis inhibitor, had no effect on tumor growth in lean mice (Figure 2E).

### 2.3. The E0771 Tumor Growth-Promoting Effect of Adipose Tissue Was Associated with Increased Tumor Lipogenesis

To determine whether tumors growing in direct contact with adipose tissue (orthotopic) metabolize lipids and glucose differently to ectopic tumors, we administrated an Intralipid emulsion containing [^3^H]-labeled triolein and [^14^C]-labeled glucose (p.o.) to E0771 tumor-bearing female mice and harvested tumor tissue for chloroform/methanol extraction. We found that the total lipid uptake was higher in orthotopic than in ectopic tumors and a similar trend was seen also for total glucose uptake (Figure 3A). This difference in total lipid uptake was due to the increased storage of neutral lipids in orthotopic tumors, as judged by the [^3^H]–triolein counts in the organic phase. De novo lipogenesis, as judged by the [^14^C]–glucose counts in the organic phase, was approximately three-fold higher in the orthotopic compared to the ectopic tumors (Figure 3B). The [^3^H]–triolein and the [^14^C]–glucose counts in the aqueous phase were, however, similar between orthotopic and ectopic tumors, indicating that catabolic processes such as fatty acid oxidation are not altered by the tumor’s anatomical localization (Figure 3C). The mRNA expression of sterol regulatory element-1 (*Srebp1*), the master regulator of lipogenesis, and acetyl-CoA carboxylase 1 (*Acaca*), which catalyzes the rate-limiting step in fatty acid synthesis, were similar between groups (data not shown). However, the fatty acid synthase (*Fasn*) mRNA levels were higher in orthotopic compared to ectopic tumors (Figure 3D).

### 2.4. Adipose Tissue Co-Culture Led to Increased Oxygen Consumption and Reduced Glycolysis in E0771 Cells

To better characterize the metabolic interactions between E0771 breast cancer and adipose tissue, we conducted a series of co-culture experiments where we compared E0771 cultured alone or co-cultured with mammary adipose tissue. Adipose tissue co-culture led to ~70% increased mitochondrial basal and ~90% increased maximal respiration. There was also a slight increase (~45%) in uncoupled respiration, while the ATP-linked respiration was more increased (~75%) in co-cultured E0771 cells compare to their controls (Figure 4A). Along with the increased respiration, glycolysis was ~20% reduced in co-cultured E0771 cells, as judged by the extracellular acidification rate (Figure 4B). We also tested the effect of adipose tissue co-culture on fatty acid oxidation-linked respiration in E0771 cells. In the absence of exogenous fatty acids, there was no effect of CPT1 inhibition on respiration in either co-cultured or control E0771 cells, suggesting that there is no or very little oxidation of endogenous long-chain fatty acids in these cells in vitro. However, co-cultured E0771 displayed reduced basal respiration, suggesting that co-culture makes E0771 more vulnerable to low-glucose conditions (Figure 4C). Treatment with BSA-conjugated palmitate triggered a CPT1-dependent increase in basal fatty acid oxidation in control E0771 cells, but not in co-cultured E0771 (Figure 4D). This indicates that the exogenous fatty acid oxidation of E0771 cells is reduced by adipose tissue co-culture. The difference in basal respiration rate between BSA and BSA–palmitate (without Etomoxir, non-mitochondrial respiration subtracted) in control E0771 was, however, insignificant (0.26 ± 0.03 vs. 0.33 ± 0.04 pmol/min/µg, *p* = 0.2), along with a trend for only slightly increased uncoupled respiration in BSA–palmitate-treated cells (0.05 ± 0.005 vs. 0.07 ± 0.005 pmol/min/µg, *p* = 0.06). This implies that palmitate supplementation increased the fatty acid oxidation at the expense of glucose oxidation, rather than increasing the total oxygen consumption. Moreover, the FCCP-stimulated respiration was similar between groups, relatively low and insensitive to CPT1 inhibition at all conditions, suggesting that FCCP-induced bioenergetic stress does not trigger increased fatty acid oxidation in E0771 cells. It should also be noted that the respiration during this experimental setup quickly reduced with time (Figure 4C,D), indicating that E0771 cells are sensitive to low-glucose conditions.

### 2.5. Adipose Tissue Co-Culture or Lactate Pretreatment Led to Increased De Novo Lipogenesis from Glucose in E0771 Cells

In line with our observations in vivo, mammary adipose tissue co-culture increased the de novo lipogenesis from glucose in E0771 cells by approximately 20% (Figure 5A). Adipose tissue is a significant producer of lactate and generates lactate from glucose even under normoxia [33,34,35,36,37]. In support of this notion, we found that the lactate production increased during 3T3-L1 adipocyte differentiation (Figure 5B) and that ex-vivo-cultured mammary and gonadal adipocytes from lean and HFD-induced obese mice produced lactate (Figure 5C). Obesity was associated with elevated serum lactate levels and enhanced the lactate production from gonadal, but not from mammary, adipocytes (Figure 5C,D). Thus, malignant breast cancers (including E0771) both produce large amounts of lactate [38,39,40] and become exposed to exogenous lactate from the adipocyte-rich microenvironment. We hypothesize that elevated lactate levels in the orthotopic tumor microenvironment (compared to the ectopic microenvironment) lead to decreased intracellular conversion of pyruvate to lactate in E0771 cells, which in turn can increase their de novo lipogenesis from glucose through an increased oxidation of pyruvate into Acetyl-CoA, which feeds into the TCA cycle. In line with our hypothesis, 25 mM lactate pretreatment for 24 h caused a ~40% increase in de novo lipogenesis from glucose in E0771 cells (Figure 5E). To test whether E0771 tumors can metabolize lactate, we conducted a [^14^C] lactate tracer experiment in mice. Most of the radio-labeled tracer was found in the aqueous phase, but we also detected significant levels in the organic phase, indicating that exogenous lactate can be taken up and used for de novo lipogenesis in this tumor type. There was, however, no difference in lactate uptake or use between orthotopic and ectopic tumors (Figure 5F).

### 2.6. Dichotomous Effect of Pharmacological FASN Inhibition on E0771 Growth In Vivo and Viability In Vitro

Our data showing that the increased tumor growth within adipose tissue is associated with increased lipogenesis prompted us to test the effect of lipogenesis inhibition on tumor growth. For these experiments, we selected Fasnall, a commercially available selective FASN inhibitor that has been shown to exert anti-proliferative effects in several different breast cancer cell lines, including MCF-7, MDA-MB-468, BT474 and SKBR3 [25]. As expected, 24 h treatment with Fasnall reduced the viability of E0771 cells in a dose-independent manner in vitro, reaching significance at 25 µM and the viability was reduced by ~45% at 100 µM (Figure 6A). In vivo, the effect of daily intraperitoneal injections with Fasnall (15 mg/kg) was less clear. On average, it had no effect on tumor growth, with a trend towards enhancing the growth of ectopically growing tumors (Figure 6B). However, some mice appeared to benefit from the Fasnall treatment. Therefore, we divided the Fasnall-treated tumors into “responders” and “non-responders” based on whether the dissected tumor weight was lighter or heavier than the average tumor weight of the vehicle controls. Approximately half of the tumors in Fasnall-treated animals were found to be “responders”, and, within this group, orthotopic tumors grew less, with a similar trend also for ectopic tumors (Figure 6C–F). In contrast, Fasnall “non-responding” tumors displayed enhanced tumor growth, and this was especially evident in ectopically growing tumors, which, on average, were five times heavier than vehicle controls (Figure 6C–F). We could recapitulate this finding via intermittent Fasnall treatment in vitro (4 h/day for 3 days); Fasnall at a concentration of 1–10 µM enhanced the viability by ~10%, 25 µM had no effect and 50–100 µM reduced the viability by ~20–30% (Figure 6G). Thus, a too low dose of Fasnall can trigger an accelerated growth of E0771 cancer cells.

## 3. Discussion

Here, we found that the luminal B breast cancer cell line E0771 grows faster when orthotopically implanted within adipose tissue than when implanted ectopically, outside adipose tissue. This result adds further support to the notion that crosstalk between adipose tissue and tumor cells promotes tumor progression [5,6,7,8,9]. Furthermore, we show that E0771 induces lipolysis and that E0771–adipose tissue crosstalk in vivo and in vitro, as well as lactate in vitro, increases de novo lipogenesis from glucose in the cancer cells. However, monotherapy with lipolysis inhibitors failed to suppress E0771 tumor growth, and the de novo lipogenesis inhibitor Fasnall was only partly effective in the used treatment regimen.

### 3.1. Lipid Metabolism in Orthotopic E0771 Tumors—Metabolic Symbiosis with Adipose Tissue

Our data suggest that triple-negative cancer cell lines together with the luminal B E0771 cell line induce lipolysis more potently than the other examined breast cancer cell lines. Furthermore, we show that E0771 tumors can both catabolize and synthesize lipids. However, it is unlikely that fatty acid oxidation and de novo lipogenesis occur simultaneously within the same cell since malonyl-CoA, which is generated during lipogenesis, inhibits CPT1, leading to reduced mitochondrial fatty acid uptake [42]. As judged by our glucose and lipid tracer data, the enhanced growth rate of orthotopic tumors compared with ectopic tumors is associated with increased lipogenesis. We thus argue that rapidly proliferating cells are the ones primarily engaged in lipogenesis. Lipogenesis is stimulated by dietary signals but also by hormonal factors, and the regulation of lipogenesis is critically dependent on SREBF1 and PPARG [43]. Indeed, SREBF1 has been shown to be vital to cancer cell proliferation both in vitro and in vivo; likewise, PPARG has been shown to be vital in insulin-stimulated cancer cell proliferation [44,45]. Besides the upregulation of *Fasn*, there was, however, no general transcriptional upregulation of the lipogenic pathway in orthotopic compared to ectopic E0771 tumors. This suggests that the increased de novo lipogenesis in orthotopic tumors in our experimental setting was driven by another mechanism. One possibility is that this cancer type already displays a high capacity for lipogenesis and that the further increase in lipogenesis in orthotopic tumors compared to ectopic tumors primarily depends on the increased availability of lipogenic substrates. Adipocytes release high amounts of lactate (this study and [33,34,35,36]). These high lactate levels in the tumor microenvironment may lead to higher intracellular lactate levels, exerting negative feedback on lactate dehydrogenase and thereby reducing the cancer cells’ own conversion of pyruvate into lactate. Larger amounts of pyruvate may then be available for oxidization into acetyl-CoA, which in turn feeds into the TCA cycle, either resupplying the electron transport chain of the mitochondria to generate ATP or providing lipogenic substrates. In line with this hypothesis, we found that pretreatment with 25 mM lactate increased the glucose-dependent de novo lipogenesis in cultured E0771 cells. Moreover, adipose tissue co-culture increased both oxidative phosphorylation and de novo lipogenesis from glucose in E0771 cells, while glycolysis (as judged by the acidification rate) was reduced. It is thus intriguing to speculate whether adipocyte-produced lactate contributes to the observed effect of adipose tissue on tumor metabolism. Lactate itself may also be a lipogenic substrate or used as fuel, as indicated by our tracer data. A central role of lactate in metabolic symbiosis between stroma and cancer cells has been described previously. For instance, Lisanti and coworkers propose that non-malignant cells are programmed to shuttle lactate to oxygenated cancer cells, which are able to use this lactate for ATP production via oxidative phosphorylation, resulting in higher proliferative capacity [46]. Lactate may also drive tumor progression by affecting gene expression [47] and activating M2-type macrophages [40].

In vivo, during mildly fasted conditions, both orthotopic and ectopic E0771 tumors displayed a significant and similar degree of fatty acid oxidation, as judged by the [^3^H]–triolein counts in the aqueous phase, while the fatty acid oxidation in cultured E0771 exposed to low-glucose (2.5 mM) conditions appeared very low and even absent in adipose tissue-cocultured E0771 cells (possibly due to increased de novo lipogenesis during this condition), and also absent in response to bioenergetic stress (FCCP treatment). Thus, cultured E0771 cells do not upregulate fatty acid oxidation when glucose levels are low, and especially not if co-cultured with adipose tissue. Rather, cultured E0771 cells respond to starvation by reducing their overall oxygen consumption rate. We have also noted that the oxygen consumption and the response to oligomycin reduce substantially in E0771 cultured in low-serum and low-glucose conditions. One possibility is that cultured E0771 cancer cells prioritize the use of their acetyl-CoA for de novo lipogenesis rather than for ATP production. These data on fatty acid oxidation illustrate that in vitro and in vivo conditions can be substantially different for cancer cells and, in this case, lead to different results.

### 3.2. Lipolysis and Lipogenesis as Therapeutic Targets in Breast Cancer Treatment

The ability of E0771 to induce lipolysis and the link between glucose-dependent de novo lipogenesis and E0771 growth prompted us to test the effect of lipolysis (Acipimox and Atglistatin) and de novo lipogenesis (Fasnall) inhibitors on tumor growth. However, neither Acipimox nor Atglistatin suppressed tumor growth, and Fasnall was, on average, equally ineffective; Fasnall suppressed tumor growth in around half of the tumors, while the other half instead displayed enhanced growth. The ineffectiveness of lipolysis inhibition on E0771 growth may be explained by the high capacity for de novo lipogenesis in E0771 cells. In contrast, the unexpected negative result from lipogenesis inhibition may be due to the increased uptake of exogenous fatty acid in Fasnall-treated cells [25], which in turn can enhance tumor progression [32]. Thus, if the concentration of Fasnall is not high enough to induce cell death, it may lead to undesired metabolic adaptations in the surviving cell. While our breast cancer model has the advantage of having tumors growing in a disease-relevant microenvironment and an intact immune system, our study has limitations. Mouse models may not completely resemble the human situation and our model did not develop metastasis during the time course of our studies. We also acknowledge that our Acipimox, Atglistatin and Fasnall treatment regimens may have affected aspects that we did not study, such as malignant transformation and/or sensitivity to conventional therapies rather than tumor growth.

In conclusion, we believe that targeting lipid- and lactate-related metabolic symbiosis between adipose tissue and cancer cells holds promise as a treatment option for breast cancer types that respond poorly to currently available therapies. However, targeting lipolysis or de novo lipogenesis separately appears rather ineffective and possibly even unsafe. Instead, we believe that it is necessary to aggressively target several metabolic pathways simultaneously, together with conventional breast cancer therapy.

## 4. Materials and Methods

### 4.1. Animals

Female C57BL/6 mice, obtained from Charles River Laboratories (Cologne, Germany), were allowed to acclimatize for one week upon arrival. The animals were maintained under standard housing conditions of a 12 h light/dark cycle and temperature with ad libitum access to water and food, regular chow or high-fat diet (60% kcal from fat, D12492 from Research Diets, New Brunswick, NJ, USA) as indicated, at the Laboratory for Experimental Biomedicine, Sahlgrenska Academy, University of Gothenburg. All experiments were performed with the permission of the Gothenburg Animal Ethics Committee.

### 4.2. E0771 Breast Cancer Model

First, 16–20-week-old female C57BL/6 mice were transplanted with the mammary breast cancer cell line E0771 suspended in equal volume Matrigel and PBS (Matrigel Basement Membrane Matrix, Corning Inc, New York, NY, USA). The transplants (1.5 × 10^5^ cells in 50 μL) were positioned either orthotopically (inguinal/mammary fat pad) or ectopically (subcutaneous, dorsolateral). The transplants were allowed to settle for 72 h, after which the mice were carefully monitored for palpation. Tumor growth was measured with a digital caliper (AgnTho’s AB, Lidingö, Sweden). Notably, this model did not develop visible metastasis during the timeframe of our studies.

### 4.3. Lipolysis Inhibitor Treatment In Vivo

To study the role of lipolysis in vivo, mice were given two different lipolysis inhibitors: Acipimox or Atglistatin. Acipimox is a niacin derivate that suppresses cyclic adenosine monophosphate (cAMP), leading to a general inhibition of lipolysis [48]. Atglistatin is a selective and competitive inhibitor of ATGL, leaving other lipases unaffected [49]. Mice received either vehicle, Acipimox (0.5 g/L, Sigma-Aldrich, St. Louis, MO, USA) in drinking water or Atglistatin (0.1 μmol/g, Cayman Chemical Company, Ann Arbor, MI, USA) by oral gavage for 14 days after E0771 transplantation. In the Acipimox study, the drinking water was replaced two times per week and the water consumption of each cage was monitored every 24 h.

### 4.4. Cell Lines and Cell Culture

The preadipocyte cell line 3T3-L1 (Zen-Bio, Research Triangle, NC, USA) was maintained in supplemented DMEM growth media containing high glucose (4500 mgL^–1^; Life Technologies, Thermo Fisher Scientific, Gothenburg, Sweden), fetal bovine serum (10%, FBS Gold; PAA laboratories) and penicillin–streptomycin (1%, Life Technologies, Thermo Fisher Scientific, Gothenburg, Sweden) [50]. Differentiation into adipocytes was carried out according to established procedures [51]. In brief, upon reaching sufficient confluency, cells were treated with a differentiation cocktail (1 μM dexamethasone, 850 nM insulin and 0.5 mM 3-isobutyl-1-methylxanthine (IBMX), for 2 days. Thereafter, the medium was interchanged with fresh medium containing a second differentiation cocktail (850 nM insulin), for an additional 2 days, after which the medium was returned to regular supplemented DMEM. The 3T3-L1 adipocytes’ maturity was determined by the prevalence of lipid droplets and then the cells were assayed, usually between 8 and 10 days from the start of differentiation.

The mammary breast cancer cell line E0771, derived from a C57BL/6 mouse [31], was maintained in normal-glucose RPMI (2000 mg L^–1^) supplemented with 10% fetal bovine serum, 1% penicillin–streptomycin, 10 mM HEPES and 1 mM sodium pyruvate.

### 4.5. Conditioned Medium Preparation

Breast cancer cell lines of luminal, HER2 and basal subtypes were grown in supplemented DMEM in 175 cm^2^ flasks for 24 h; thereafter, the medium was replaced with serum-free DMEM for 24 h. The resulting supernatant was sterile-filtered (0.2 μm) and concentrated with a 3 kDa molecular weight cutoff centrifugal filter (Macrosep Advance Centrifugal Device, Pall Life Sciences, Port Washington, NY, USA), and the cell count was determined in order to pre-dilute the samples representing 1 × 10^6^ million cells/mL media. In brief, the centrifugal filter was washed twice with dH_2_O (20 mL, 20 min 5000 rpm) to remove residual glycerol; thereafter, conditioned medium was added and concentrated down to 1 mL (1 h, 5000 rpm, 4 °C). The concentrate was washed with fresh serum-free DMEM (10 mL, 1 h, 5000 rpm, 4 °C). The final concentrate was diluted based on cell concentrate.

### 4.6. Lipolysis In Vitro

First, 3T3-L1 adipocytes or pre-adipocytes were treated with either regular or conditioned media (24 h), after which the supernatant was harvested to measure free glycerol using the Free Glycerol Reagent, according to manufacturer’s protocol (Sigma-Aldrich, St. Louis, MO, USA).

### 4.7. Lipid, Glucose and Lactate Metabolism In Vivo

Radio-labeled triolein, [9,10-^3^H(N)], was prepared by evaporation in a conical glass vial with nitrogen gas; thereafter, Intralipid (20%, soybean oil, egg phospholipids and glycerin) and glucose, D-[^14^C(U)], were added. The solution was sonicated (20 s, three times) and stored on ice. In a separate experiment, lactate L-[^14^C(U)] was used alone. Animals were fasted (4 h) prior to the oral administration of tracer (2 µCi for each tracer/mouse and 60 mg/mouse of TG 20%). The exact amount of tracer given to each animal was determined by weighing the syringe and needle before and after administration. Mice were euthanized and tumor tissues were harvested 2 h after tracer administration. Tissues were homogenized using a Tissue Lyser II (Qiagen, Hilden, Germany) in a chloroform:methanol solution (1 mL, 2:1) and stored overnight (4 °C). Thereafter, 0.5 mL 1 M CaCl_2_ was added prior to centrifugation (3000 rpm, 4 °C, 20 min). The aqueous and organic phases were transferred to separate scintillation vials and the ^3^H and ^14^C counts per minute (CPM) of all samples were normalized to the exact amount of tracer given to each animal (based on tracer density and syringe weight difference before and after gavage) and to the dissected tissue weight.

### 4.8. RNA Isolation, cDNA Synthesis and Quantitative Real-Time PCR

RNA from tissue and cell lysates was isolated and purified using the ReliaPrep RNA Cell MiniPrep System (Promega Corporation, Fitchburg, WI, USA), according to the manufacturer’s protocol, except for the processing of adipose tissue, which required the removal of lipids prior to chloroform extraction. The concentration of the total RNA was determined by a NanoDrop, and cDNA was generated through reverse transcription (500–1000 ng RNA) using a mixed priming strategy, according to the manufacturer’s protocol (qScript Flex cDNA Synthesis Kit, Quanta Biosciences, Beverly, MA, USA). The gene expression was quantified through qRT-PCR using SYBR green (Fast SYBR^®^ Green Master Mix, Applied Biosystems, Waltham, MA, USA) and the relative relative ΔC_t_ method normalized to β-actin (Actb). Primers were used at a concentration of 0.5 μM.

### 4.9. E0771 and Adipose Tissue Co-Culture Studies

E0771 breast cancer cells were seeded in 12-well companion plates (Corning Inc, New York, NY, USA) at a density of 0.4 × 10^6^ cells per well and were allowed to adhere overnight. The following day, whole inguinal/mammary adipose tissue was dissected (including removal of inguinal lymph nodes) and kept in sterile PBS supplemented with 5 mM glucose. The tissues were further manually cut into 5–10 mg fragments in a petri dish. Thereafter, samples were filtered (100 µm mesh) and washed with PBS and placed in cell culture inserts (0.4 μm PET, Corning Inc, New York, NY, USA) containing 35–70 mg tissue per insert. The system was then incubated for 24 h in RPMI media, as described previously, with the exception of the addition of 2.5 nM dexamethasone and 0.5 nM human insulin to better accommodate the adipose tissue, as adapted from a protocol for the organ culture of human adipose tissue [52]. E0771 cells used as controls were cultured in the same conditions, thus also receiving 2.5 nM dexamethasone and 0.5 nM human insulin for 24 h.

### 4.10. Characterization of Mitochondrial Function and Glycolysis

E0771 cells from the co-culture experiment, including control cells, were trypsinated, allowing for the pooling of cells from six wells. Cells were then redistributed in Seahorse XF96 cell culture microplates (at 100,000 cells/well) and allowed to adhere for 10 h before running the Seahorse Cell Mito Stress, Glycolysis or Fatty Acid Oxidation Test assays using a Seahorse XFe96 Analyser (Agilent Technologies, Santa Clara, CA, USA).

Oxidative phosphorylation: The Seahorse Cell Mito Stress Test (Agilent Technologies, Santa Clara, CA, USA) was performed according to the manufacturer’s protocol. In brief, cells were incubated in assay media for 1 h (Seahorse XF assay medium supplemented with 25 mM glucose and 1 mM sodium pyruvate, pH 7.4) prior to exposure to a series of chemicals inhibiting mitochondrial oxidative phosphorylation, ATPase inhibitor oligomycin, proton uncoupler FCCP and complex I and III inhibitor rotenone/antimycin A, added in the respective order. Oligomycin and FCCP were titrated as a part of the assay optimization, to 2 μM and 0.5 μM, respectively. The obtained OCR values in response to the different inhibitors were used to calculate basal, maximal, non-mitochondrial respiration, ATP production, coupling efficiency, proton leak and spare respiratory capacity. All Mito Stress Test assays were repeated three times.

Aerobic glycolysis: Glycolytic function was determined by the Seahorse Cell Glycolysis Stress Test assay (Agilent Technologies, Santa Clara, CA, USA). Glycolysis, glycolytic capacity, glycolytic reserve and non-glycolytic acidification were thus determined by measurements of the extracellular acidification rate (ECAR), according to the manufacturer’s protocol. In brief, cells were incubated in assay media for 1 h (Seahorse XF assay medium supplemented with 1 mM glutamine, pH 7.4) prior to exposure to a series of chemicals affecting glycolysis, glucose (10 mM), oligomycin (1 μM) and 2-deoxyglucose (50 mM), added in the respective order. All glycolysis stress test assays were repeated three times.

Fatty acid oxidation: Fatty acid oxidation was measured by the XF Fatty Acid Oxidation assay combined with the Cell Mito Stress Test (Agilent Technologies, Santa Clara, CA, USA) to determine whether fatty acid oxidation contributed to the measured OCR. This assay was performed according to the manufacturer’s protocol, with some exceptions. In brief, it is recommended to starve cells overnight in the Seahorse cell culture microplates before performing this assay; however, E0771 cells (and most cancer cell lines) have a rapid metabolism and are very sensitive to prolonged culture without replenishing media, and thus the onset of starvation occurs quickly. This is associated with low respiration and almost no response to any compounds targeting the mitochondrial respiration, especially oligomycin. Hence, on the day of the assay, cells were only incubated for 45 min in fatty acid oxidation (FAO) assay buffer (and not starved in the substrate limited buffer) prior to the addition of the carnitine palmitoyltransferase-1 (CPT-1) inhibitor Etomoxir (40 μM). Thereafter, BSA or palmitate:BSA was added, after which the Mito Stress Test was conducted as previously described. Endogenous and exogenous fatty oxidation were determined by comparing OCR in, respectively, the BSA control in the presence or absence of etomoxir and palmitate: BSA in the presence or absence of etomoxir. All fatty acid oxidation assays were repeated three times.

All OCR and ECAR values were normalized to total protein content as determined by the Pierce BCA Protein Assay Kit (Thermo Fisher Scientific, Waltham, MA, USA).

### 4.11. De Novo Lipogenesis In Vitro

E0771 cells were seeded at a density of 100,000 cells/well in a 12-well plate and were serum-starved in normal-glucose RPMI for 1 h prior to the addition of radio-labeled glucose (D-[3-^3^H(U)], 50 μCi/well). Cells were incubated at RT for 10 min; thereafter, excess radio-labeled tracer was washed away with PBS, and cells were lysed by addition of NaOH (0.5 mL, 50 mM). CPM was measured in the organic phase as above.

### 4.12. Lipogenesis Inhibitor Treatment

The role of de novo lipogenesis was studied both in vitro and in vivo by treatment with the selective fatty acid synthase (FASN) inhibitor Fasnall benzenesulfonate salt (Cayman Chemical Company, Ann Arbor, MI, USA). In vivo, mice were transplanted with E0771 and given daily intraperitoneal injections with either vehicle or Fasnall (15 mg/kg) for approximately three weeks, after which animals were euthanized and tissues harvested. In vitro, E0771 breast cancer cells were seeded at a density of 15,000 cells/cm^2^, were allowed to adhere overnight and were treated with Fasnall benzenesulfonate salt in the presence of 10% fetal bovine serum. Viability was measured with the Crystal Violet method [53]. Briefly, wells were washed two times with PBS and incubated with 0.1% Crystal Violet solution (Sigma-Aldrich, MERCK, Darmstadt, Germany) for 15 min at room temperature. After thorough washing of unbound dye with dH2O, bounded dye was dissolved with 33% acetic acid. Absorbance was read on a SpectraMax i3x plate reader (Molecular Devices, San Jose, CA, USA) at 540 nm and 670 nm.

### 4.13. Statistics

GraphPad Prism 9 (GraphPad Software, San Diego, CA, USA) was used for statistical analysis. Values are shown as means ± standard error of the mean (SEM). Comparisons were performed using two-way ANOVA or two-tailed *t*-test, log transformation was performed as necessary to achieve normal distribution and *p* < 0.05 was considered statistically significant.

## Figures and Tables

**Figure 1 ijms-22-11881-f001:**
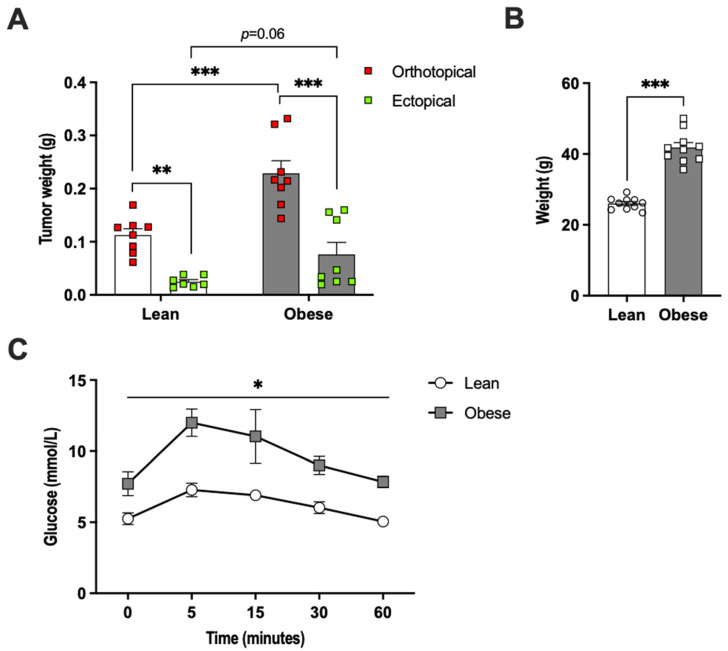
(**A**) E0771 tumor and (**B**) body weights two weeks post-implantation in lean and high-fat-diet-induced obese female mice (N = 8–10/group). Panel (**C**) shows an oral glucose tolerance test performed prior to E0771 implantation (N = 4–6/group). Data are presented as mean ± SEM; * *p* < 0.05, ** *p* < 0.01 and *** *p* < 0.001.

**Figure 2 ijms-22-11881-f002:**
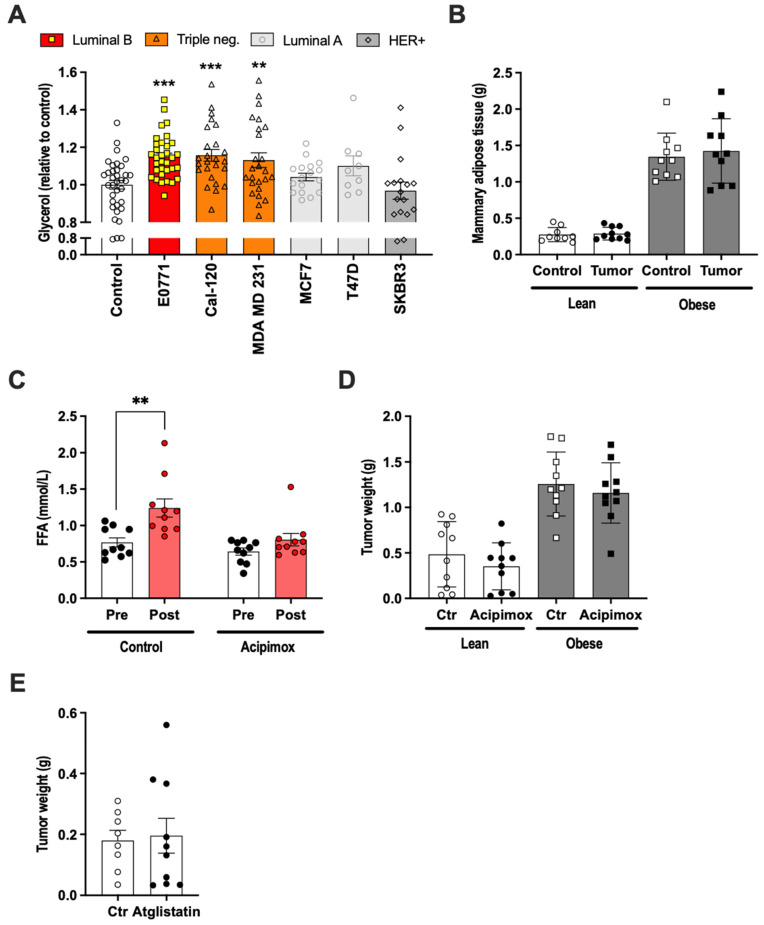
(**A**) Glycerol released from 3T3-L1 adipocytes treated with conditioned media (representing media from 1 × 10^6^ million cells/mL media) from breast cancer cell lines of different subtypes (N = 9–35/group). (**B**) Control and tumor-associated mammary adipose tissue weight in lean and high-fat-diet-induced obese female mice two weeks after orthotopic E0771 implantation (N = 10/group). (**C**) Serum free fatty acids levels in lean female mice before and two weeks after orthotopic E0771 implantation with and without Acipimox treatment (N = 10/group). (**D**) Tumor weights in control and Acipimox-treated lean and high-fat-diet-induced obese female mice two weeks after orthotopic E0771 implantation (N = 10/group). (**E**) Tumor weights in control and Atglistatin-treated lean female mice two weeks after orthotopic E0771 implantation (N = 8–10/group). Data are presented as mean ± SEM; ** *p* < 0.01 and *** *p* < 0.001.

**Figure 3 ijms-22-11881-f003:**
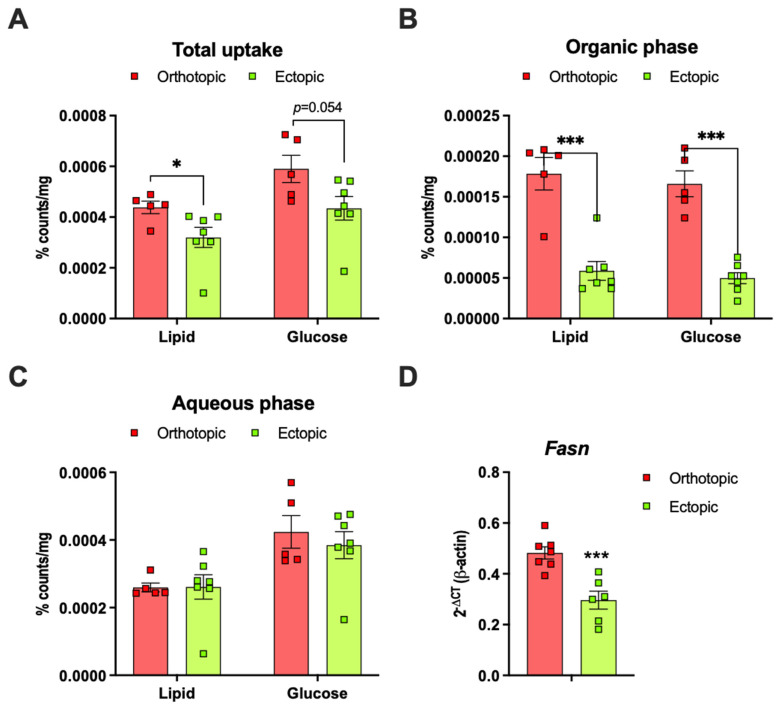
(**A**) Total lipid and glucose uptake in orthotopic and ectopic E0771 tumors from lean female mice two hours after an oral load of Intralipid emulsion containing [^3^H]–triolein and [^14^C]–glucose. The lipid ([^3^H]–triolein) and glucose ([^14^C]–glucose) counts in the (**B**) organic and the (**C**) aqueous phase of E0771 tumors (N = 5–7/group). (**D**) Fatty acid synthase (*Fasn*) mRNA expression in orthotopic and ectopic E0771 tumors from lean female mice two weeks after E0771 implantation (N = 6–7/group). Data are presented as mean ± SEM; * *p* < 0.05 and *** *p* < 0.001.

**Figure 4 ijms-22-11881-f004:**
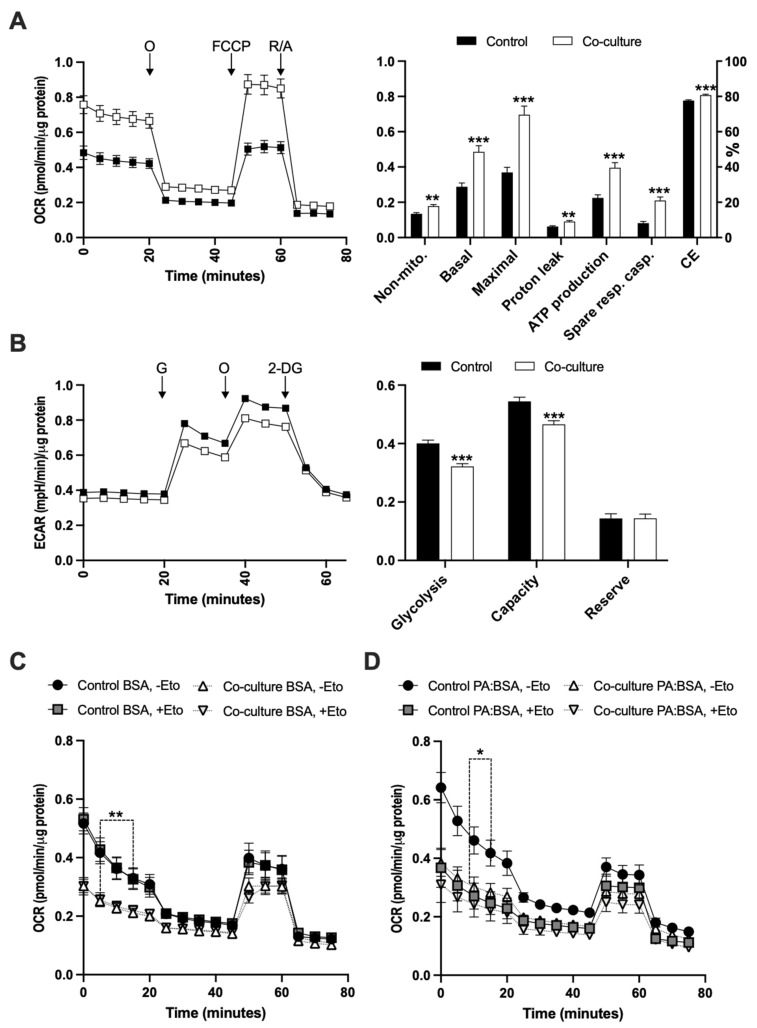
(**A**) Mitochondrial function as judged by changes in non-mitochondrial respiration, basal respiration, maximal respiration, proton leak-related respiration, ATP production-linked respiration, spare respiratory capacity and coupling efficiency (CE) determined from the oxygen consumption rate (OCR) in response to subsequent addition of oligomycin (O), FCCP and rotenone (R)/antimycin A (A) as indicated in the left panel of control and 24 h mammary adipose tissue co-cultured E0771 cells. Data are presented as OCR normalized to total protein levels (N = 30/group). (**B**) Glycolytic function, as determined by basal glycolysis rate, glycolytic capacity and glycolytic reserve, was estimated from the extracellular acidification rate (ECAR) of control and 24 h mammary adipose tissue co-cultured E0771 cells in response to subsequent addition of glucose (G), oligomycin (O) and 2-Deoxy-D-glucose (2-DG) as indicated in the left panel. Data are presented as ECAR normalized to total protein levels (N = 30/group). (**C**) Endogenous and (**D**) exogenous fatty acid oxidation estimated from the OCR of control and 24 h mammary adipose tissue co-cultured E0771 cells under low-glucose conditions (1.5 mM glucose, with or without the carnitine palmitoyltransferase-1 (CPT-1) inhibitor Etomoxir (eto), with or without BSA-conjugated palmitate (PA:BSA) in response to subsequent addition of oligomycin (O), FCCP and rotenone (R)/antimycin A (A) as indicated in the left panel A (N = 12/group). Data are presented as mean ± SEM; * *p* < 0.05, ** *p* < 0.01 and ****p* < 0.001.

**Figure 5 ijms-22-11881-f005:**
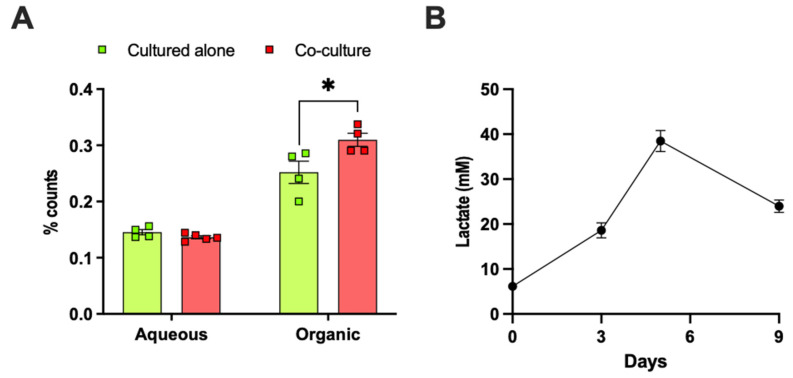
(**A**) [^14^C] counts in aqueous and organic phase (representing de novo lipogenesis) in control and 24 h mammary adipose tissue co-cultured E0771 cells in response to 10 min [^14^C]–glucose treatment at a glucose concentration of 11.1 mM (N = 6/group, * *p* < 0.05 for cultured alone vs. adipose tissue co-culture). Lactate levels in (**B**) 24 h culture media from differentiating 3T3-L1 adipocytes at the indicated time points (0 days = preadipocytes; 8 days = mature adipocytes) (N = 10/group) and in (**C**) culture media from isolated mammary (MA) and gonadal adipocytes (GA) (N = 6–10/group) cultured with the so-called MAAC method [41] (# *p* < 0.05, ## *p* < 0.01 for lean vs. obese GA; * *p* < 0.05 for obese MA vs. obese GA), and in (**D**) serum from lean (N = 7) and high-fat-diet-induced obese (N = 6) female mice (* *p* < 0.05 for lean vs. obese). Notably, the serum lactate levels are higher than expected; this may reflect that the sera came from terminal blood obtained under deep isoflurane anesthesia. (**E**) [^14^C] counts in aqueous and organic phase in control and 24 h 25 mM lactate-pretreated E0771 cells in response to 15 min [^14^C]–glucose treatment at a glucose concentration of 11.1 mM (N = 4–6/group, ** *p* < 0.01 for control vs. 25 mM lactate pretreatment). (**F**) [^14^C] counts in aqueous and organic phase in orthotopic and ectopic E0771 tumors from lean female mice two hours after an oral load with a [^14^C] lactate tracer (N = 5/group). Data are presented as mean ± SEM.

**Figure 6 ijms-22-11881-f006:**
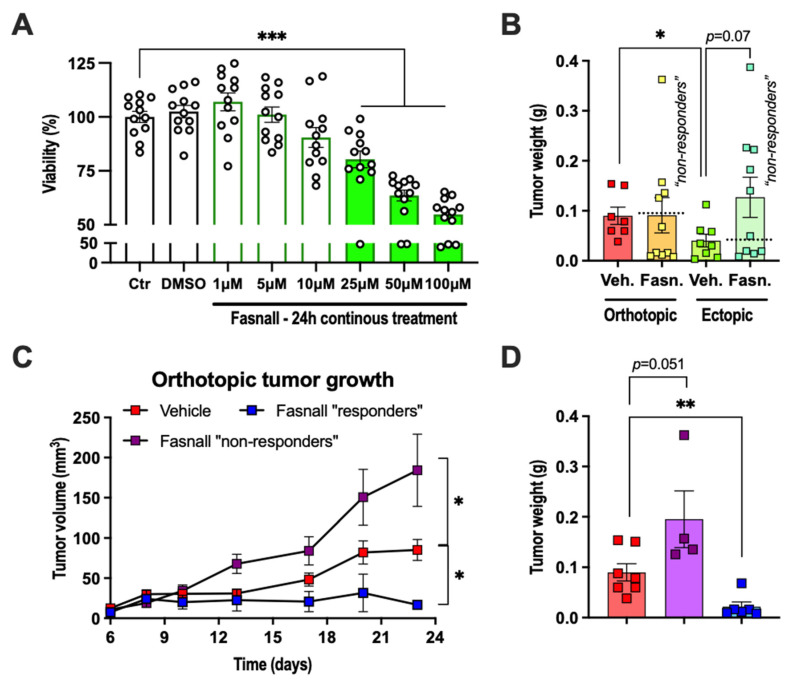
(**A**) Viability of cultured E0771 cells after 24 h continuous treatment with Fasnall at the indicated concentrations (N = 12/group). (**B**) Orthotopic and ectopic E0771 tumor weights after 23 days of vehicle or Fasnall treatment (15 mg/kg, daily i.p. injections, N = 7–10/group, * *p* < 0.05 for the orthotopic vs. ectopic tumors in the vehicle group). The dashed lines indicate the division between Fasnall “non-responders” and “responders”. (**C**,**D**) Orthotopic and (**E**,**F**) ectopic E0771 tumor growth curves and tumor weights after 23 days of Fasnall treatment comparing vehicle with Fasnall “non-responders” and “responders” separately (* *p* < 0.05, ** *p* < 0.01 and *** *p* < 0.001 for vehicle vs. Fasnall). (**G**) Viability of cultured E0771 cells after three days of intermittent treatment (4 h/day) with Fasnall at the indicated concentrations (N = 10–12/group, ** *p* < 0.01 and *** *p* < 0.001 for control vs. Fasnall). Data are presented as mean ± SEM.

## Data Availability

Data is contained within the article.

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
