# Peer review of "Adipose Tissue—Breast Cancer Crosstalk Leads to Increased Tumor Lipogenesis Associated with Enhanced Tumor Growth"

_ijms, 2021, doi:10.3390/ijms222111881_

Round 1

Reviewer 1 Report

The manuscript i well written, the results are clearly presented and the conclusion well founded. 

Minor points: in fig 1C the curve symbols are barely discernible

Author Response

Reply to Reviewer 1

The manuscript i well written, the results are clearly presented and the conclusion well founded. 

Minor points: in fig 1C the curve symbols are barely discernible

Response: Thank you for your positive evaluation of our work. We have now changed the symbols of Fig. 1C and we have also made all the figures larger to make them more legible as per request of another reviewer.

Reviewer 2 Report

Dear authors, I have reviewed your manuscript entitled: ‘Adipose tissue - breast cancer crosstalk leads to increased tumor lipogenesis associated with enhanced tumor growth’ and have the following comments:

  1. I my opinion the first paragraph of introduction should be re-organized. Authors should provide information related to future outcomes in respect to triple negative BrC and than describe luminal B BrC.
  2. From my point of view adipose tissue enhances tumor progression not only in a paracrine manner also in auto- and endocrine manner. Authors should mention about leptin and adiponectin hormones-derived from adipose tissue, which stimulate tumor growth and invasion.
  3. The figures should be re-designed since they are hardly legible.
  4. In figure 1 the description of lean and obese is similar, thus is hard to differentiate, please change the legend.
  5. In last paragraph of discussion Authors should provide strength and limitation of the study

Author Response

Reply to Reviewer 2

Dear authors, I have reviewed your manuscript entitled: ‘Adipose tissue - breast cancer crosstalk leads to increased tumor lipogenesis associated with enhanced tumor growth’ and have the following comments:

Response: Thank you for your positive evaluation of our work as well as your constructive comments that we believe have helped us to significantly improve our manuscript. Below is our response to each of your comments separately.

  1. I my opinion the first paragraph of introduction should be re-organized. Authors should provide information related to future outcomes in respect to triple negative BrC and than describe luminal B BrC.

Response: We agree with you; the first paragraph should have been more informative and balanced, and we have therefore replaced the first paragraph with a new introductory paragraph:

“Breast cancer is a common cause of death among women, but earlier detection and better treatments have greatly improved the chances of survival [1]. However, the prognosis is dependent both on the stage and on the molecular subtype. There are four main molecular subtypes of breast cancer: 1) luminal A (estrogen receptor-positive and/or progesterone receptor-positive [ER+ and/or PR+] and negative for human epidermal growth factor receptor 2 [HER2–]), 2) luminal B (ER+ and/or PR+/HER2+), 3) HER2-enriched (ER– and PR–/HER2+) and 4) triple-negative (ER–, PR– and HER2–). Luminal A breast cancers are the most common, and these cancers typically respond well to hormonal therapy and show the best prognosis [2]. In contrast, triple negative cancers show the worst prognosis, but also luminal B and non-luminal HER2+ subtypes have been associated with unfavorable prognosis [2-4]. Thus, there is still a need for new treatments to further improve the outcome for women with breast cancer.”

  1. From my point of view adipose tissue enhances tumor progression not only in a paracrine manner also in auto- and endocrine manner. Authors should mention about leptin and adiponectin hormones-derived from adipose tissue, which stimulate tumor growth and invasion.

Response: We completely agree with you and have modified the introduction to further clarify that adipose tissue can enhance tumor progression also in an endocrine manner. Paragraph 2 of the introduction now reads (red text is new):

”A growing body of research shows that adipose tissue enhances tumor progression in a paracrine and endocrine manner through many different plausible mechanisms [5-9]. For instance, the adipocyte-derived hormones leptin and adiponectin have been shown to affect tumor progression [10, 11].

However, this study focuses on the local paracrine crosstalk as we compare tumors growing within adipose tissue with tumors growing outside adipose tissue and both these tumors will be exposed to adipose tissue-derived hormones (although tumors within adipose tissue most likely will be exposed to a higher concentration of these hormones, but then we would define such impact as paracrine)

  1. The figures should be re-designed since they are hardly legible.

Response: We agree, the figures and fonts are too small. We have now made new figures that hopefully are more legible.

  1. In figure 1 the description of lean and obese is similar, thus is hard to differentiate, please change the legend.

Response: Thank you for pointing this out; we have now changed the symbols and legend.

  1. In last paragraph of discussion Authors should provide strength and limitation of the study

Response: We have now expanded this last paragraph, adding the following:

“While our breast cancer model has the advantage of having tumors growing in a disease-relevant microenvironment and an intact immune system, our study has limitations. Mouse models may not completely resemble the human situation and our model does not develop metastasis during the time course of our studies. We also acknowledge that our Acipimox, Atglistatin and Fasnall treatment regimens may have affected aspects that we did not study such as malignant transformation and/or sensitivity to conventional therapies rather than tumor growth.”

References

  1. Winters, S.; Martin, C.; Murphy, D.; Shokar, N. K., Breast Cancer Epidemiology, Prevention, and Screening. Prog Mol Biol Transl 2017, 151, 1-32.
  2. Fallahpour, S.; Navaneelan, T.; De, P.; Borgo, A., Breast cancer survival by molecular subtype: a population-based analysis of cancer registry data. CMAJ open 2017, 5, (3), E734-E739.
  3. Jaaskelainen, A.; Roininen, N.; Karihtala, P.; Jukkola, A., High Parity Predicts Poor Outcomes in Patients With Luminal B-Like (HER2 Negative) Early Breast Cancer: A Prospective Finnish Single-Center Study. Front Oncol 2020, 10, 1470.
  4. Ades, F.; Zardavas, D.; Bozovic-Spasojevic, I.; Pugliano, L.; Fumagalli, D.; de Azambuja, E.; Viale, G.; Sotiriou, C.; Piccart, M., Luminal B breast cancer: molecular characterization, clinical management, and future perspectives. Journal of clinical oncology : official journal of the American Society of Clinical Oncology 2014,32, (25), 2794-803.
  5. Carter, J. C.; Church, F. C., Mature breast adipocytes promote breast cancer cell motility. Exp Mol Pathol 2012,92, (3), 312-7.
  6. Dirat, B.; Bochet, L.; Dabek, M.; Daviaud, D.; Dauvillier, S.; Majed, B.; Wang, Y. Y.; Meulle, A.; Salles, B.; Le Gonidec, S.; Garrido, I.; Escourrou, G.; Valet, P.; Muller, C., Cancer-associated adipocytes exhibit an activated phenotype and contribute to breast cancer invasion. Cancer research 2011, 71, (7), 2455-65.
  7. Tan, J.; Buache, E.; Chenard, M. P.; Dali-Youcef, N.; Rio, M. C., Adipocyte is a non-trivial, dynamic partner of breast cancer cells. Int J Dev Biol 2011, 55, (7-9), 851-9.
  8. Wang, Y. Y.; Lehuede, C.; Laurent, V.; Dirat, B.; Dauvillier, S.; Bochet, L.; Le Gonidec, S.; Escourrou, G.; Valet, P.; Muller, C., Adipose tissue and breast epithelial cells: a dangerous dynamic duo in breast cancer. Cancer letters 2012, 324, (2), 142-51.
  9. Lengyel, E.; Makowski, L.; DiGiovanni, J.; Kolonin, M. G., Cancer as a Matter of Fat: The Crosstalk between Adipose Tissue and Tumors. Trends Cancer 2018, 4, (5), 374-384.
  10. Parida, S.; Siddharth, S.; Sharma, D., Adiponectin, Obesity, and Cancer: Clash of the Bigwigs in Health and Disease. International journal of molecular sciences 2019, 20, (10).
  11. Lin, T. C.; Hsiao, M., Leptin and Cancer: Updated Functional Roles in Carcinogenesis, Therapeutic Niches, and Developments. International journal of molecular sciences 2021, 22, (6).